# Cost-Effectiveness Analysis of an Intracranial Stereotactic Radiotherapy Service for Brain Metastasis in a North Queensland Regional Cancer Centre

**DOI:** 10.3390/cancers18010163

**Published:** 2026-01-02

**Authors:** Qichen Zhang, Lan Gao, Neha Das, Timothy Squire, Daniel Stoker, Reshma Shakya, Deepti Patel, Abhishek Joshi, Tao Xing

**Affiliations:** 1Department of Radiation Oncology, Townsville University Hospital, 100 Angus Smith Drive Douglas, Townsville, QLD 4814, Australia; qichen.zhang@health.qld.gov.au (Q.Z.);; 2Greater Brisbane Clinical School, Faculty of Medicine, The University of Queensland, Brisbane, QLD 4072, Australia; 3Deakin Health Economics, Institute for Health Transformation, Faculty of Health, Deakin University, Burwood, VIC 3125, Australia; 4College of Medicine and Dentistry, James Cook University, Townsville, QLD 4814, Australia; 5Department of Medical Oncology, Townsville University Hospital, Townsville, QLD 4814, Australia

**Keywords:** stereotactic radiosurgery, brain metastasis, health economy, radiotherapy

## Abstract

Rural and regional Australian patients, especially Aboriginal and Torres Strait Islander patients, are faced with multifaceted challenges when receiving a referral to metropolitan centres for specialist medical care, which affects the uptake of recommended treatment and therefore negatively impacts the outcome. Being able to access specialist care closer to home improves the accessibility and timeliness of recommended treatment. To the best of our knowledge, this is the first study reporting the cost-effectiveness of the implementation of an intracranial SRS service at an Australian regional cancer centre. This study provides evidence to initiate further discussions on the identification of suitable cancer care models to deliver specialist care from funding and policy support perspectives.

## 1. Introduction

Brain metastasis is a significant cause of morbidity and mortality among cancer patients. Up to 40% of patients with solid malignancies develop brain metastases [1]. Cancer Council Queensland statistics reported 5378 cancer diagnoses in the Far North Queensland, North Queensland, and Mackay regions in 2022 [2], which means that more than 2000 patients could potentially require care for brain metastases at some point during their cancer treatment. The incidence is rising with the increased detection of clinically occult disease through staging magnetic resonance scans (MRI) [3] and increased overall survival due to the improved efficacy of systemic therapies [4].

Stereotactic radiotherapy has proven effectiveness in controlling the disease in the brain with minimal toxicity in randomised controlled trials (RCTs) and large cohort studies [5,6,7,8]. Stereotactic radiosurgery (SRS) has become the standard of care and an essential treatment for brain metastases, with a 12-month local control rate of 72% and better preserved cognition and quality of life, with improved survival benefits in certain subgroups of patients, including patients with single brain metastases [9] and patients younger than 50 years [10]. Patients from rural Australia, however, face challenges in access this specialised radiotherapy treatment close to home, as it is usually offered in high-volume metropolitan centres as a statewide service in many Australian jurisdictions, mainly due to the expertise and resources required to provide it [11,12]. Reports from the National Rural Health Alliance identified that not only is the combined incidence of all cancers highest in regional Australia, but cancer mortality rates are also higher in rural and remote populations [13]. The reason for poorer cancer outcomes in rural Australian populations is multifactorial, and delays or inability to access care due to geographical challenges play an important role. Data from Queensland suggested that 40% of rural cancer patients were not able to commence treatment within the recommended timeframe [14].

In Queensland, prior to the implementation of an SRS service at our regional centre, North Queensland patients had to travel to metropolitan tertiary centres to receive this SRS treatment, with the closest being Brisbane. SRS can be delivered in a single fraction, or a hypofractionated schedule can be used for larger lesions or lesions in eloquent areas over three to five fractions [5,7,8]. It usually takes approximately 2 weeks for category 1 patients to be seen after a referral is made to allow time for case discussion in multidisciplinary meetings and clinic scheduling. The initial consult can be carried out either in person or via telehealth. Another 1–2 weeks is usually required until treatment planning and delivery. Patients then will need to travel to Brisbane to stay for 1–7 days depending on dose/fractionation selection. The stay could be longer if further care or observation is needed, whether due to side effects or other concurrent medical issues. If patients require further SRS later on, the same process will be repeated. Due to logistic and financial barriers, a lack of social support and limited health literacy, patients from regional, rural and remote regions, and especially Aboriginal and Torres Strait Islander patients, quite often miss the opportunity to receive this treatment. This was anecdotally noticed through clinical practice prior to the implementation of the SRS service.

Many patients require multiple SRS treatment courses during their cancer journey due to distant brain failure, with the development of new lesions outside the previous SRS field. This occurs in 36% of patients, as reported in a previous retrospective analysis [8]. Access to SRS treatment locally allows patients and their families to save significant funds via travel, accommodation, time off work and other intangible costs—for example, emotional distress due to being away from home and family. To address this unmet need, our regional hospital implemented the first North Queensland intracranial SRS service in 2022.

The provision of SRS in regional hospitals warrants rigorous evaluation due to its potential to address disparities in access to advanced cancer treatment. Patients residing in regional and remote areas frequently encounter substantial geographic, financial and logistical barriers when accessing SRS at metropolitan centres, including long-distance travel, extended accommodation stays and associated caregiver burdens. These barriers may delay treatment initiation, reduce compliance with recommended follow-up and exacerbate inequities in cancer outcomes. The local delivery of SRS could mitigate these challenges, enhance patient convenience and reduce non-medical costs, while maintaining comparable clinical outcomes to metropolitan care. Evaluating the economic implications of local SRS delivery is essential to guide health service planning, resource allocation and policy decisions aimed at improving equitable cancer care access.

## 2. Materials and Methods

### 2.1. Study Population

We conducted a prospective study at our institution after the implementation of an intracranial stereotactic radiotherapy service in September 2022. We included all patients treated with stereotactic radiotherapy from September 2022 to December 2024 at our centre for limited intracranial metastatic disease. The study was approved by the local institutional human research ethics committee. Data were collected and stored on the statewide electronic medical record system. We recorded patient demographic details, types of primary malignancy, details of previous treatment, including systemic therapy and surgery, and radiotherapy dose/fractionation. Treatment outcomes are also reported, including the local control rate, incidence of radiation necrosis and overall survival.

When the service was first established, only patients with solitary brain metastasis were considered for treatment at our regional centre. Treatment was delivered to intact lesions or surgical cavities postoperatively. Other institutional criteria to offer treatment included a maximum lesion size < 40 mm, a minimum lesion size > 8 mm, a maximum of any single planning treatment volume (PTV) ≤ 10 cm^3^ for intact lesions or ≤15 cm^3^ for cavities and an Eastern Cooperative Oncology Group performance status (ECOG) 0–2. Patients with multiple intracranial metastases or metastases close to critical organs (e.g., brainstem) were referred to metropolitan centres. Prior to treatment, all cases were discussed in the unit chart round and with the medical physics department to ensure safety to treat at our centre. Absolute contraindications were an inability to lie still safely for planning and treatment, diffuse leptomeningeal involvement and pregnancy. Relative contraindications warranting additional caution when considering treatment were an inability to undergo brain MRI, severe claustrophobia, >1 cm midline shift on imaging, uncontrolled seizures, concurrent use of radio-sensitising medications and connective tissue disorders.

### 2.2. SRS Services

At our centre, treatment planning was performed using the Monaco planning system (Elekta, Stockholm, Sweden) and patients were treated on an Elekta Versa HD™ linear accelerator. Planning CT is acquired with a 1 mm slice thickness and co-registered with diagnostic MR brain. Planning MR is occasionally used and obtained on an Elekta Unity™ MR-LINAC system. For intact lesions, the gross tumour volume (GTV) is delineated based on planning CT and the post-contrast T1-weighted sequence of MRI, and the clinical target volume (CTV) is equal to the GTV. For surgical cavities, CTV_cavity includes the resection cavity as determined on available imaging and any suspicious residual disease, including the surgical tract. For both intact lesion and cavity treatment, the PTV consists of isotropic expansion of a minimum of 2 mm from the CTV. The prescription dose fraction is decided based on the tumour volume, tumour size, previous radiotherapy and dose to organs at risk (OAR). We usually suggest withholding chemotherapy for seven days, with targeted therapy for one day prior to brain SRS. The decision is made in discussion with medical oncology colleagues. After the completion of treatment, patients are followed up with via MR brain at three monthly intervals or according to symptomatology.

### 2.3. Follow-Up

Patients receive initial follow-up at 6 weeks post-treatment. Subsequently, patients are followed up with via MR brain at 3-month intervals or at the time of symptomatic progression. Side effect profiles, including the occurrence of radiation necrosis, are recorded at follow-up appointments.

### 2.4. Cost Analysis

The cost analysis adopted a societal perspective, incorporating both medical and non-medical costs. Medical costs included the cost of radiotherapy (including follow-up appointments), recorded prospectively for each study participant. Costs for imaging studies, such as MRI scans required for treatment or follow-up, were included. It was assumed that the costs for radiotherapy and diagnostic tests were identical at the regional and metropolitan hospitals. Other medical costs unrelated to brain metastasis lesions were not considered in the analysis. Costs were compared within the same patient cohort, with the control arm representing the counterfactual scenario in which all participants underwent radiotherapy at a metropolitan centre. The non-medical costs included travel, accommodation and informal caregiver costs. Upfront capital, including equipment and training, and ongoing operational costs are similar for metropolitan and regional centres and therefore not considered to contribute to the cost difference in this analysis.

#### 2.4.1. Travel Costs

Patients’ postcodes were used to determine the travel itinerary and associated costs, based on the mode(s) of transport. Costs included taxi fares to the local airport, airfares to the treatment destination (where applicable) and taxi fares from destination airports to the treatment hospital. Similar costs were applied for the return journey from the hospital. We assumed that one caregiver accompanied the patient; therefore, their airfare was also included. Unit costs for flights and taxis (per kilometre travelled) were obtained from publicly available sources. If multiple courses of travel were required during the treatment planning and delivery processes, including for consultation and diagnostic tests, the cost for each episode of travel was calculated.

#### 2.4.2. Accommodation Costs

Accommodation costs were based on the unit cost of a double-bed hotel room. Patients travelling by air to the treatment destination were assumed to require accommodation. For treatment at the regional hospital, accommodation was assumed necessary if the patient’s residence was more than 100 km from the hospital. Length of stay was assumed to be two nights for metropolitan centre treatment and seven nights for regional hospital treatment, based on observations through clinical practice.

#### 2.4.3. Informal Care Costs

The opportunity cost of informal care was calculated by multiplying the number of workdays lost by the average Australian in weekly total earnings (pro-rated to a daily rate). One caregiver was assumed per patient. For metropolitan centre treatment, caregivers were assumed to lose three days of paid work; for regional hospital treatment, this was eight days. Earnings data were sourced from the Australian Bureau of Statistics [15].

Travel and accommodation costs were based on values from 2025, and caregiver costs were adjusted to corresponding 2025 values, using a consumer price index. Cost sources are included in Appendix A.

### 2.5. Statistical Analysis

For each patient, total costs were calculated as the sum of medical and non-medical components. Mean costs along with 95% confidence intervals (CI) are reported for both the regional hospital and metropolitan centre scenarios. We did not conduct statistical comparisons between groups, as the costs for the comparative scenario were based on counterfactual rather than observed data.

A Kaplan–Meier survival analysis was performed on the study cohort to estimate overall survival following radiotherapy. Time zero (baseline) for time-to-event analyses was defined as the date of the first day of radiotherapy. The primary event was death from all causes, and patients who were alive at the last follow-up was censored at that date. The Kaplan–Meier method was employed to estimate cumulative survival probabilities over time, and survival distributions were summarised with median survival times and corresponding 95% confidence intervals where applicable.

Similarly, the time to radiation necrosis was analysed using a Kaplan–Meier curve for the study cohort; one patient was excluded due to the last follow-up date being earlier than the documented date of necrosis, rendering time-to-event calculation invalid. For this analysis, time zero was again defined as the date of radiotherapy, with radiation necrosis considered the event of interest. Patients without necrosis were censored at the last available follow-up date. This analysis enabled the estimation of the cumulative incidence of radiation necrosis over time, allowing for an assessment of treatment-related adverse effects. The analysis was run on the entire cohort of 34 patients.

The clinical outcomes of the control group were derived from a literature search of studies [7,8] reporting the clinical outcomes of cancer patients with brain metastasis treated with SRS, including one study performed at an Australian metropolitan tertiary hospital [8]. All analyses were conducted using StataCorp. 2023. Stata Statistical Software: Release 18. College Station, TX, USA: StataCorp LLC.

## 3. Results

### 3.1. Patient Characteristics

Between September 2022 and December 2024, 34 consecutive patients received intracranial SRS for either intact brain metastases or postoperative resection cavities (Table 1). The median age was 65 years (range: 49–78). The most common primary malignancy was non-small cell lung cancer, accounting for 42% of the patients (15/34). Nearly half underwent adjuvant SRS to the surgical cavity following resection. Concurrent systemic therapy was uncommon is this cohort (chemotherapy 6%, *n* = 2; immunotherapy 12%, *n* = 4). The most frequently used regimen was 24 Gy in three fractions (*n* = 24, 71%).

### 3.2. Treatment Costs

The mean total non-medical and medical cost was AUD 7450 (95% CI 6657–8254) in the state capital of Brisbane and AUD 6690 (95% CI 6005–7375) at our regional hospital (Table 2). Mean medical costs were equivalent, whereas non-medical costs were lower locally. At the regional centre, the mean per-patient non-medical costs were AUD 436 for travel, AUD 499 for accommodation and AUD 747 for informal care—each lower than the corresponding costs at the metropolitan centre.

### 3.3. Survival Outcomes

The median survival time following radiotherapy was 15.7 months (95% CI: 7.8–23.6) (Figure 1). Radiation necrosis occurred in eight patients (24%); the median time to event was not estimable (Table 3). Progressive intracranial disease developed in 56% (*n* = 19) during follow-up. Published Canadian and metropolitan Australian series have reported medial survival durations of 9.6 and 11.8 months and radiation necrosis rates of 13.2% and 15.6%, respectively. These comparable outcomes suggest that brain SRS delivered at our regional centre achieves similar effectiveness and safety to that in established metropolitan services. 

## 4. Discussion

To the best of our knowledge, this is the first study evaluating the cost-effectiveness of implementing an intracranial stereotactic radiosurgery program at a regional health service in Australia. In addition to the well-recognised benefits of receiving medical treatment closer to home—such as increased convenience and improved continuity of care—local specialist radiotherapy services have the potential to reduce the financial burden on both patients and the healthcare system.

Although there is limited brain metastasis-specific evidence regarding the accessibility of specialist brain SRS services for rural and regional patients, or about the impact of accessibility on outcomes, general healthcare trends indicate that patients from rural and regional communities—particularly Aboriginal and Torres Strait Islander people—face greater barriers to accessing specialised treatment, such as brain SRS, than their metropolitan counterparts [16]. Rural Australians usually wait longer for specialist care, including cancer care [16,17], and experience poorer health outcomes, as reflected in various health performance indicators, such as shorter life expectancy and higher mortality rates [16].

Implementing intracranial SRS services within regional health services and hospitals is one potential approach to addressing disparities in access to this essential oncological treatment between regional and metropolitan settings. Evidence from other healthcare disciplines suggests that providing specialist care locally improves access [18], enables family support, enhances quality of life during treatment [19] and encourages treatment uptake [20]. For Aboriginal and Torres Strait Islander patients, receiving treatment in or closer to their own region also enables the provision of culturally appropriate care and reduces the psychosocial burden associated with referrals to distant metropolitan centres [21,22,23]. One important outcome to include in future analysis is patient-reported outcomes to better support the psychosocial benefits of receiving treatment locally.

Our study highlights that potential for regional intracranial SRS services to reduce the financial burden on patients, their families and the healthcare system. While medical expenses are similar between regional and metropolitan centres, regional delivery substantially reduces non-medical costs associated with travel, accommodation and informal caregiving. In Queensland, many of these costs are covered by the Patient Travel Subsidy Scheme (PTSS) [24]; therefore, reducing such costs can enable PTSS funding to support more patients in accessing specialist care when travel is unavoidable.

We also prospectively collected outcome data to demonstrate that, with careful patient selection to identify patients that are suitable for management with the level of expertise available locally, intracranial SRS can be safely delivered at regional centres with comparable outcomes to urban quaternary centres. Our regional centre collaborates with its metropolitan counterparts in the form of multidisciplinary meetings, discussions and consultations to determine the best approach for individual patients and support them through it.

Establishing regional specialist services is not without challenges. A key advantage of centralised specialist services is the ability to maintain high patient volumes, which maintains staff proficiency, optimises resource utilisation, reduces per-patient costs and has been shown to improve patient outcomes, especially in surgical settings [25,26], although existing studies to support this are limited by confounding factors and biases [27]. Nonetheless, our study shows that it is feasible to build local expertise while maintaining cost-efficiency. The concerns related to low patient volumes can be mitigated through strong partnerships with high-volume centres, regular staff training for skill maintenance and the adoption of collaborative service models such as the hub-and-spoke model [28]. This model formalises the relationship between metropolitan and rural hospitals, enabling shared learning, case discussions and the application of high-volume centre experiences to regional settings. Universally adopted clinical practice guidelines and cross-institutional quality assurance (QA) can also help to establish benchmarks for service provision, holding treatment delivered across different institutions to comparable quality standards. Telehealth technologies, such as remote contouring and system commissioning, may also be used to utilise urban resources in regional settings where needed.

Since the implementation of our intracranial SRS program, we have continued to work closely with our local multidisciplinary teams, including medical physics, radiotherapy, medical oncology, neurosurgery and other relevant disciplines. We also strengthened our collaborations with metropolitan colleagues. Each patient is carefully assessed to determine whether treatment can be safely provided locally or whether referral to a metropolitan centre is more appropriate, ensuring that both clinical and non-clinical needs are met.

The clinical outcomes observed in our cohort are comparable to those reported in historical series [5,6,7] and recent data from Peter MacCallum Cancer Centre [8]. However, only patients with solitary lesions that were not within the eloquent areas of the brain were treated locally. Ongoing follow-up and continuous collaboration with metropolitan institutions are necessary to ensure that the rural patient cohort is assigned a treatment plan that would offer the maximum benefits to suit both their clinical and personal needs. Another limitation of our study is the small sample size. As we continue to provide services and follow-up for our patients, we will be able to conduct further cost-effectiveness analyses and report on patient outcomes across a larger cohort and over a longer period to benchmark them against metropolitan centres. Our partnerships with urban cancer centres and multidisciplinary efforts will also continue to benefit rural and regional patients in North Queensland. Additionally, since we compared the two cohorts based on a counterfactual scenario, our study provides a path for future research including and analysing patients in the two cohorts.

Another emerging area is the shift to preoperative SRS over postoperative SRS for brain metastases. Evidence suggests that preoperative SRS can improve local control and reduce the risk of leptomeningeal disease [29,30,31]. However, adopting this treatment protocol might require additional funding and resources—particularly in health services with limited or no neurosurgery capabilities. Even for regional health services with a neurosurgery capacity, existing capacity constraints often make expansion challenging, given the large geographical areas and populations that they serve. Our institution is the only centre in North Queensland that provides neurosurgical services, serving a large catchment area with a population of 700,000 people, which includes Aboriginal and Torres Strait Islander patients, as well as very remote communities. More resources would be required to provide timely and state-of-the-art management of intracranial metastases for all patients. Policy-level advocacy is needed to facilitate further multidisciplinary discussion and ensure adequate support for the implementation of such advanced services [20].

## 5. Conclusions

Implementing regional intracranial SRS services represents a potentially cost-effective strategy to improve access to this essential oncological treatment for rural and regional Australians. Close collaboration with metropolitan centres and robust multidisciplinary support are critical in ensuring appropriate patient selection and optimal care delivery. Further benchmarking, policy engagement and funding advocacy will be essential to align regional service development with national and state healthcare priorities.

## Figures and Tables

**Figure 1 cancers-18-00163-f001:**
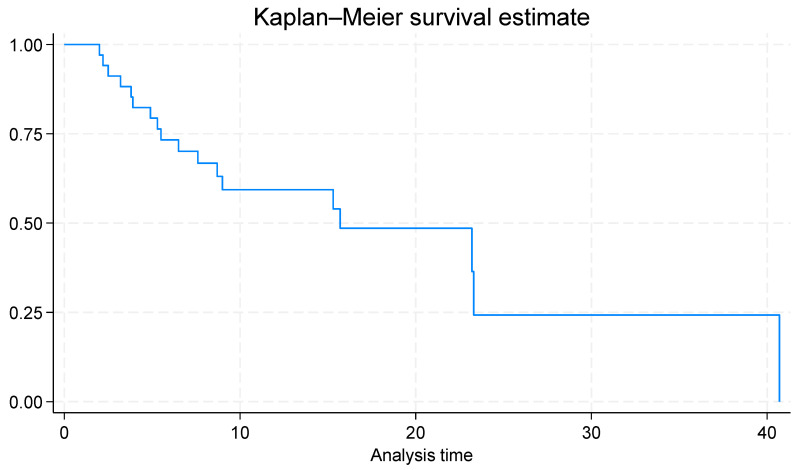
Kaplan–Meier survival estimates of overall survival.

**Table 1 cancers-18-00163-t001:** Baseline characteristics of the study population.

Parameter	Data
Sample size	34
Age (mean, range)	65 (Range 49–78)
Type of cancer (*n*, %)	Non-small cell lung cancer—15 (42%) Melanoma—5 (14%) Gastrointestinal—3 (9%) Small cell lung cancer—2 (6%) Renal cell carcinoma—1 (3%) Breast—2 (6%) Others—6 (17%)
Concurrent chemotherapy (*n*, %)	2 (6%)
Concurrent immunotherapy (*n*, %)	4 (12%)
Surgery (*n*, %)	16 (47%)
Radiotherapy dose (some patients received multiple Rx doses, *n*, %)	24 Gy in 3 fractions: 24 (71%) 30 Gy in 5 fractions: 5 (15%) 25 Gy in 5 fractions: 3 (9%) 27 Gy in 3 fractions: 3 (9%) 27.5 Gy in 5 fractions: 2 (6%)
Radiotherapy-related necrosis (*n*, %)	8 (24%) (grade 1 or 2 necrosis)
Progressive disease (*n*, %)	19 (56%)

**Table 2 cancers-18-00163-t002:** Results of medical and non-medical costs for two scenarios.

	Radiotherapy in Metropolitan Centre	Radiotherapy in Regional Hospital
Medical		
Mean (95% CI)	AUD 5754 (5103–6405)	AUD 5754 (5103–6405)
Non-medical		
Mean (95% CI)	AUD 2554 (2262–2846)	AUD 1682 (929–2435)
Travel		
Mean (95% CI)	AUD 1045 (883–1207)	AUD 436 (228–645)
Accommodation		
Mean (95% CI)	AUD 651 (576–726)	AUD 499 (272–726)
Informal care		
Mean (95% CI)	AUD 858 (759–957)	AUD 747 (407–1086)
Total cost		
Mean (95% CI)	AUD 7450 (6657–8254)	AUD 6690 (6005–7375)

Assumptions: -Three patients were from another regional town, so we assumed that two used flights and needed accommodation and one drove with no need for accommodation. -For regional radiotherapy, (a) for those in the North Queensland region, if the distance was >100 km, we assumed that they needed accommodation; (b) if a patient took a flight to the regional centre, then they also needed accommodation and needed a caregiver as the patient had to stay; (c) patients from the catchments of other hospital and health services took fights and needed accommodation. -The regional hospital was a regional cancer centre in North Queensland. -The metropolitan centre was a metropolitan hospital in Brisbane.

**Table 3 cancers-18-00163-t003:** Comparison of clinical outcomes with those reported from Australian metropolitan hospitals.

Study	Hospital	Sample Size	Time_0	Median OS (95% CI) in Months	Mean Age	Cancer Type	Surgery Performed	Immuno-Therapy	Previous Radiotherapy
Myrehaug et al., 2021 [7]	Sunnybrook Health Sciences Centre, Ontario, Canada	220	Radiotherapy	11.8 (9.9–14)		Breast: 45 (20%) NSCLC: 105 (47%) Melanoma: 18 (8%) RCC: 14 (6%) Gastrointestinal: 17 (7%) Other: 21 (10%)	-	-	-
Zhang et al., 2023 [8]	Peter MacCallum Cancer Centre, Melbourne, Australia	152	First day of hypofractionated stereotactic radiosurgery	9.6 (8.5–12.6)	61.9	NSCLC: 49 (32%) Breast: 38 (25%) Melanoma: 26 (17%) Other: 39 (26%)	-	23%	14%
Current study	A regional hospital in North Queensland	34	First radiotherapy	15.7 (7.8–23.6)	65	NSCLC: 15 (42%) SCLC: 2 (6%) RCC: 1 (3%) Breast: 2 (6%) Others: 6 (17%) GI: 3 (9%) Melanoma: 5 (14%)	16 (50%)	4 (11%)	-

NSCLC: non-small cell lung cancer; SCLC: small cell lung cancer; RCC: renal cell carcinoma; GI: gastrointestinal; OS: overall survival; CI: confidence interval.

## Data Availability

Raw data were generated at Townsville University Hospital. Derived data supporting the findings of this study are available from the corresponding author, T.X., on request.

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
