# Peer review of "Cost-Effectiveness Analysis of an Intracranial Stereotactic Radiotherapy Service for Brain Metastasis in a North Queensland Regional Cancer Centre"

_cancers, 2026, doi:10.3390/cancers18010163_

Round 1

Reviewer 1 Report

Comments and Suggestions for Authors

The paper focused on the cost effettiveness of the brain SRS in a specific rural area, where patients need to travel for receveing RT treatment; also, since the area is vast and sparsely populated, the patients and their caregivers also need accomodation for allowing the treatment to be delivered. Since we are talking about brain SRS and the authors pointed out that the patients underwent brain  MRI before the treatment (even because the MRI is used for planning and delivering the treatment) , I wonder if the necessity of undergoing a MRI with constrast might not impact on the cost of the treatmnent as well.

In Europe the ncesessity of a Brain MRI before brain SRS and SBRT is a burden for clinicians, due to the ffact that the patients requiring the MRI outnumbered the facility avaliable (esoeciali for thin slices MRI).

However, the authors did not include this as part as financial burden for patients, since the MRI is usually done 1 or 2 weeks before the simulation CT so patiented need to travel for undergoimng MRI as well. 

I thinkl it should be included in the analyses, considering the sparsely populated area in which the study population is located (and we can guess that the MRI are avalialble in the same hospital facilities where the radiotherapy units are)

Author Response

Comments 1: 

"The paper focused on the cost effettiveness of the brain SRS in a specific rural area, where patients need to travel for receveing RT treatment; also, since the area is vast and sparsely populated, the patients and their caregivers also need accomodation for allowing the treatment to be delivered. Since we are talking about brain SRS and the authors pointed out that the patients underwent brain  MRI before the treatment (even because the MRI is used for planning and delivering the treatment) , I wonder if the necessity of undergoing a MRI with constrast might not impact on the cost of the treatmnent as well.

In Europe the ncesessity of a Brain MRI before brain SRS and SBRT is a burden for clinicians, due to the ffact that the patients requiring the MRI outnumbered the facility avaliable (esoeciali for thin slices MRI).

However, the authors did not include this as part as financial burden for patients, since the MRI is usually done 1 or 2 weeks before the simulation CT so patiented need to travel for undergoimng MRI as well. 

I thinkl it should be included in the analyses, considering the sparsely populated area in which the study population is located (and we can guess that the MRI are avalialble in the same hospital facilities where the radiotherapy units are)"

Response 1:

Thank you very much for this comment. We completely agree that the cost of MR and cost associated with travelling for MR can significantly contribute to the overall medical cost of treatment. 

In Australia, medical imaging required for diagnosis and treatment can usually be performed within reasonable timeline. The cost of imaging is fully covered by the universal health insurance scheme and would incur no additional cost to patients. At institutional and health insurances levels, the costs for MR imaging are the same for metropolitan and regional services. This cost has been included in 'medical cost' category, including the cost for follow-up appointments, for radiotherapy delivered at both metropolitan and regional hospitals. (Section 2.4 and 3.2). Travel costs are fully reimbursed under the patient travel subsidy scheme in Australia as well. Travel cost for imaging studies at societal level has been accounted for under the general 'travel cost' category. 

Following this comment, it has now been added to section 2.4 that imaging costs have been considered when calculating medical costs (line 171-173), and travel cost for imaging or other reasons has been consider when calculating travel cost (line 186-188).

Thank you very much again for this suggestion!

Reviewer 2 Report

Comments and Suggestions for Authors

This well written paper argues for the cost-effectiveness of establishing a local stereotactic radiosurgery (SRS) service for brain metastases in a regional Australian cancer centre, as a means to overcome access barriers for rural and remote patients.

Key Argument:

Local SRS reduces non-medical costs significantly: The mean total cost per course was A$6,690 locally vs. A$7,450 in Brisbane, driven largely by savings in travel, accommodation, and informal caregiving costs.

Clinical outcomes are comparable: With careful patient selection (solitary, non-eloquent metastases), regional centres can deliver safe and effective SRS, achieving median survival of 15.7 months and a radiation necrosis rate of 24%, both within expected standards.

Societal and patient-centered benefits: Keeping care local reduces travel burden, caregiver strain, and psychological distress for patients and families, which is especially important for rural, remote, and Indigenous populations.

The study concludes that regional SRS is a viable, cost-saving model that improves equity in cancer care access. This is an importan finding, which merits attention and can help in arguing for an improvement of logistical availability of SRS in remore regions. I feel that the paper contains crucial informations.

The non-medical cost reductions is demonstrated in this paper, though somewhat obvious: still important to have exact data on  the primary economic benefit, which is might be compelling for policymakers. In terms of patient-centere perspective, this paper adresses the psychosocial burdens of travel and dislocation. In addition, the authors make the case that regional SRS can meet clinical standards with existing infrastructure and expertise.

However, there are some weaknesses and some  gaps:

  1. Missing Infrastructure & Recurring Cost Analysis:

The paper does not account for the upfront capital (equipment, training) and ongoing operational costs of establishing and maintaining an SRS unit.

The low patient volume (n=34) raises questions about whether the per-patient savings justify the fixed costs of a regional unit.

A comparison with high-volume metropolitan centres—which may achieve economies of scale—is absent, weakening the cost-effectiveness claim.

  1. Limited Clinical Outcome Reporting Could Be Condensed:

Much of the clinical results section confirms expected standards of care, which could be summarized more succinctly to focus on the cost and accessibility argument.

Only straightforward cases were treated locally; complex cases were referred out. This may overstate the feasibility and outcomes of a full-scope regional service.

Recommendations for Strengthening the paper

Include a budget-impact analysis that incorporates capital investment, staffing, maintenance, and training costs.

Compare per-patient costs at low-volume vs. high-volume centres to address economies of scale.

Shorten the clinical outcomes section to a high-level summary, emphasizing that standards of care were met.

Incorporate patient-reported outcomes or qualitative insights to better support the psychosocial benefits of local care.

.

Overall Assessment

This paper makes a compelling initial case for regional SRS based on patient accessibility and non-medical cost savings, but its cost-effectiveness argument is incomplete without analysis of setup and maintenance expenses. The clinical results, while reassuring, could be condensed to sharpen the focus on economic and equity-driven policy recommendations. I recommend the acceptance of this paper, if the minor revision are met.

Author Response

Comments 1:

"1. Missing Infrastructure & Recurring Cost Analysis:

The paper does not account for the upfront capital (equipment, training) and ongoing operational costs of establishing and maintaining an SRS unit.

Thank you very much for the comment. One of the discussion point of our paper that we would like to highlight is indeed that our brain SRS service was implemented using existing resources and infrastructure (Abstract line 36-37, line 56-58). The upfront capital for equipment has been accounted for by the existing equipment in service, leading to no additional costs on commissioning. While additional training is required, this cost is absorbed as part of the professional development and continuing education which is an expected costs. Similarly, operational costs has not directly incurred as extra cost given the operational resources used are already in place to account for all services provided by the department. The per treatment operational expenses are expected to reduce over time upon further service provision, and the upfront capital and operational costs would also be similar for both metropolitan and regional centres in longer term, which would not contribute significantly to the cost analysis in this study. It has been added to section 2.4 now that upfront capitals and operational costs have been considered following this comment. (Line 178-180) 

The low patient volume (n=34) raises questions about whether the per-patient savings justify the fixed costs of a regional unit.

Thank you for this comment and we agree that one of the limitations of this study is the sample size. One of the main reasons that the sample size is small is that treatment at our centre is limited to solitary brain metastases at current stage. As the reviewer commented as well, the role of this paper is to provide exact data to support discussions with stakeholders in policy making. With ongoing service provision at our hospital, we will be able to report the outcome of larger cohorts on longer-term follow-up.  

A comparison with high-volume metropolitan centres—which may achieve economies of scale—is absent, weakening the cost-effectiveness claim."

Thank you very much for the comment and agree this is worth considering. Literatures to support economics of scale in healthcare have been reviewed, and a discussion point has now been added to the Discussion section. Line 324-327. There is evidence to support that high-volume centres may have the advantages of better patient outcome and reduces per-patient cost. Though it has been pointed out by a recent review (Bhattarai, N.; McMeekin, P.; Price, C.; Vale, L. Economic evaluations on centralisation of specialised healthcare services: a systematic review of methods. Health Economic Research. BMJ Open. 2016. 6:e011214. DOI: 10.1135/bmjopen-2016-011214.) that evidence in this space is confounding and biased due to the complex nature of assessing health services. 

Comments 2:

"Limited Clinical Outcome Reporting Could Be Condensed:

Much of the clinical results section confirms expected standards of care, which could be summarized more succinctly to focus on the cost and accessibility argument.

Thank you for this comment. The outcome section has been revised to be more succinct.

Only straightforward cases were treated locally; complex cases were referred out. This may overstate the feasibility and outcomes of a full-scope regional service."

We agree with this comment and identify this as one of the limitations of this analysis. We aim to expand the service to accomodate the care needs of our patients locally as much as possible. The costs associated with treatment far away from home not only includes financial costs, but also psychosocial costs which would not be possible to quantify, line 291-299. We hope the conclusion of paper serve as a discussion starter when deciding what the best mode is to deliver care for rural and regional patients, accepting a full-scope regional service may not be the most cost-effective model and centralised care would be required for complex cases. 

Recommendations for Strengthening the paper

Include a budget-impact analysis that incorporates capital investment, staffing, maintenance, and training costs.

Thank you for the recommendation. This has been responded above and specified in section 2.4. 

Compare per-patient costs at low-volume vs. high-volume centres to address economies of scale.

Further literature review and discussion point added (line 317-319), acknowledging this comparison would be limited by compounding factors and biases (Bhattarai 2025). Per capital costs are listed and compared in Table 2 and Supplement 1. 

Shorten the clinical outcomes section to a high-level summary, emphasizing that standards of care were met.

Outcome section has now been re-written and shortened.

Incorporate patient-reported outcomes or qualitative insights to better support the psychosocial benefits of local care.

We collected patient reported outcome using EQ-5D-5L questionnaire during this study. However, we didn't generate any reportable data due to the small patient cohort and low completion rate. We would be able to add this to future publications once the results are available. It has been added to discussion as a limitation with the need to consider patient reported quality of life highlighted (Line 299-300).  

Thank you very much for the insightful comments. We have improved our manuscript based on the recommendations and would address some of the recommendations in our future work as well when the data become available with longer term service provision and follow-up.